# Characterization of the Key Aroma Compounds in Different Yeast Proteins by GC-MS/O, Sensory Evaluation, and E-Nose

**DOI:** 10.3390/foods12163136

**Published:** 2023-08-21

**Authors:** Jiahui Chen, Dandan Pu, Yige Shi, Baoguo Sun, Hui Guo, Ku Li, Yuyu Zhang

**Affiliations:** 1China Key Laboratory of Geriatric Nutrition and Health, Beijing Technology and Business University, Ministry of Education, Beijing 100048, China; cjh16636184361@163.com (J.C.); 18518351472@163.com (D.P.); jennyshiyige@163.com (Y.S.); sunbg@btbu.edu.cn (B.S.); 2Food Laboratory of Zhongyuan, Beijing Technology and Business University, Beijing 100048, China; 3Key Laboratory of Flavor Science of China General Chamber of Commerce, Beijing Technology and Business University, Beijing 100048, China; 4Hubei Provincial Key Laboratory of Yeast Function, 168 Chengdong Road, Yichang 443003, China; guohui@angelyeast.com (H.G.); liku@angelyeast.com (K.L.)

**Keywords:** yeast proteins (YPs), electronic nose, aroma compounds, addition experiments

## Abstract

The unique odors of yeast proteins (YPs) are decisive for their application in meat substitutes. Sensory evaluation, electronic nose, and gas chromatography–mass spectrometry/olfactory (GC-MS/O) were combined to characterize the aroma profiles and aroma-active compounds of YPs. The sensory evaluation results indicate that the sweaty aroma had the strongest intensity in YP #10, followed by rice bran, sour, and plastic. The electronic nose could effectively distinguish the aroma differences among five YPs. A total of 27 aroma-active compounds in the five YPs were identified by GC-MS/O. The concentration of 2-methyl-propanoic acid (6.37 μg/kg), butanoic acid (47.46 μg/kg), 3-methyl-butanoic acid (22.50 μg/kg), and indole (943.40 μg/kg) in YP #10’s aroma was higher than that of the other YPs. The partial least squares regression method results show that o-cresol, (3S)-3,7-dimethyloct-7-en-1-ol, benzyl alcohol, octanal, 2-methyl-propanoic acid, butanoic acid, 3-methyl-butanoic acid, hexanal, heptanal, and indole were predicted as the potential aroma-active compounds significantly contributing to the aroma profiles of the five YPs. Addition experiments confirmed that the overall aroma profile intensities of the five YP samples were extended with the addition of these ten compounds, verifying their significant contributions.

## 1. Introduction

Yeast is a single-celled organism. In early times, yeast was used to ferment flour and make wine. During bread preparation, the important aroma compounds, including 2-phenylethanol, 1-heptanol, heptanal, benzaldehyde, phenylacetaldehyde, and ethyl octanoate, are generated by yeast fermentation in the bread crumbs [1]. Moreover, the diversity of yeast strains can affect the flavor complexity of wine [2]. Yeast cells are rich in nutrients such as protein, carbohydrates, nucleic acids, vitamins, and lipids, among which protein is the most abundant. Protein is an indispensable component in living organisms, and it is the basic organic matter that makes up cells. At present, animal protein is the main protein consumed by humans, but farming animals may have a huge impact on the environment [3,4]. Additionally, excessive intake of animal meat could increase the incidence of cardiovascular and cerebrovascular diseases. Researchers have developed several kinds of plant protein as new sustainable substitutes for meat analogs, but plant protein has lower consumer acceptance due to its unique bean aroma [5]. Since the actual demand for protein increases along with population growth, microbial proteins are also being considered due to their high production rate and low environmental pollution. Moreover, microbial protein is not only an alternative source of high-quality protein, but its other cellular components, such as lipids, carotenoids, (pro)vitamins, and essential amino acids, are also of value in the growing field of novel nutrition [6].

Yeast protein is obtained by nuclease hydrolysis, centrifugation, dispersion, and drying. Pacheco et al. determined that the average protein content of yeast protein after extraction by sodium perchlorate was 75% [7]. A net protein utilization measured result showed that the nutritional value of casein protein was more than 90%. Yeast protein also contains a fairly high content of lysine, which is used as a limiting amino acid in grains [8]. Yeast protein can be used as a nutritional additive in food to provide essential nutrients for vegetarians, athletes, and people who are deficient in B vitamins [9]. It has good functional properties, such as gelation, solubility, and stability, and could be used as a wall material for new encapsulated products [10]. The good hydration ability and oil retention ability of yeast protein make it a good filler for meat, which could improve the tenderness, juiciness, and product stability of meat products [11]. However, yeast protein has an intense characteristic yeasty odor (comprehensive sensory attributes) [12], which has affected its development. Therefore, the identification of the key aroma compounds contributing to the yeasty odor in yeast protein is an efficient way to control its unpleasant aroma, and further, to regulate its formation during fermentation and manufacturing.

Yeast products with different aroma composition variances from different manufacturing sources, processing techniques, and strain types have different volatile components. Mahadevan et al. reported that the aroma compositions of yeast extracts showed significant differences due to different manufacturers, even when using the same yeast extract batches and the same processing methods [13]. At present, a lot of work has been carried out to investigate the aroma compounds in yeast extracts by gas chromatography–mass spectrometry combined with olfactory detection (GC-MS/O), but identification of the key aroma compounds is still lacking deep investigation, resulting in the limited application of yeast protein. Zheng et al. reported the key aroma compounds, including styrene, octanal, o-xylene, furfuryl alcohol, and isovaleric acid, in yeast proteins by solid-phase microextraction (SPME) combined with gas chromatography–mass spectrometry/olfactory (GC-MS/O) and aroma extract dilution analysis (AEDA) [14]. However, most of the studies have mainly been focused on yeast extract rather than YPs. GC-MS/O combined with AEDA has been the most popular analytical method. By application of this method, Zhang et al. determined that 3-methylbutyric acid, 3-picoline, and 4-methylphenol were the main odorants in yeast extracts and further confirmed their identification accuracy by a recombination experiment [15]. Wang et al. confirmed that nonanal, γ-decalactone, dimethyl disulfide, octanal, benzoneacetaldehyde, styrene, acetophenone, 2-methoxyphenol, p-cresol, and indole were the key aroma compounds in yeast extracts by GC-O and multivariable statistics [12]. Additionally, heat treatment can increase the concentrations of furfural and dimethyl sulfide compounds in yeast extracts, increasing the aroma intensities of meaty and roasty odors according to gas chromatography–ion mobility spectrometry analysis [16]. There is a lack of research on how the application/replacement of yeast protein in vegetable protein analogs affects the flavor quality. Moreover, little research has been conducted on decoding the key aroma compounds of yeast protein, which is not conducive to the development and promotion of the microbial protein industry.

The aims of this work were to (1) determine the aroma profiles of yeast proteins by sensory evaluation and electronic nose analysis, (2) identify the aroma-active compounds in yeast proteins by headspace SPME-GC-MS/O, (3) correlate the relationship between the aroma-active compounds in the aroma profiles and screen the potential aroma compounds, (4) and confirm the key aroma-active compounds and their contributions by addition tests.

## 2. Materials and Methods

### 2.1. Materials and Chemicals

In this experiment, different types of yeast proteins, #10, #11, #12, #13, and #14, were provided by Angel Yeast Co., Ltd. (Yichang, China). Yeast protein is produced in two steps: firstly, the yeast cell walls undergo self-dissolution and fragmentation by adding enzymes; then, the yeast proteins are obtained by centrifugation, dispersion, and drying.

Helium (99.999%) was purchased from Bazhou Anxing Gas Co., Ltd. (Bazhou, China). 2-methyl-3-heptanone (99%) and methanol (chromatographically pure, 99.9%) were purchased from Beijing J&K Scientific (Beijing, China). Alkanes (C6~C30) (chromatographically pure 99.9%) were purchased from the Sigma-Aldrich Company (St. Louis, MO, USA) in the United States. Ultrapure water was purchased from the Wahaha Group (Hangzhou) Co., Ltd. (Hangzhou, China).

### 2.2. Sensory Evaluation

An aroma profile evaluation of the YPs was conducted according to our previous work, with some modifications [17]. Ten healthy panelists (5 males and 5 females, aged 24–30 years) experienced in aroma description and intensity scoring were recruited for sensory evaluation. Firstly, the panelists were trained to become familiar with and distinguish the differences among variance aromas using 54-aroma kits (Le Nez du Vin, France). This part of the training helped the panelists identify different aromas in food to describe the differences among the YPs. The aroma descriptors were determined based on expert discussions on the sensory attributes among the five YPs. Seven aroma profiles, including sour (acetic acid), rice bran, sweaty (isovaleric acid), floral (phenyl alcohol), sweet (vanillin), roasty (2,6-dimethylpyrazine), and plastic (phenol) were finally determined. Then, 1.50 g of each YP was loaded into an odorless transparent plastic bottle, numbered with 3 digits, and then presented to the panelists randomly. They were asked to score the aroma intensity on a 9-point scale from 1 (very weak) to 9 (very strong). During the evaluation, the evaluators had a rest period of 5 min after completing each group. All samples were repeated three times.

### 2.3. Electronic Nose Analysis

An electronic nose (E-nose) is an array of gas sensors that has a fingerprint response to specific volatiles. The volatile molecules react with the sensing material of the gas sensor, resulting in irreversible changes in the electrical correlation properties. These changes can be used by pattern recognition algorithms, such as artificial neural networks (ANNs), to differentiate and classify different samples [18]. A PEN3 electronic nose (Airsense Analytics GmbH, Schwerin, Germany) consisting of an array of 10 selective sensors was used to distinguish the aromas of different YPs. The instrument is sensitive to different aroma compounds; Table 1 shows the main functions of the sensors of different arrays [19]. YP powder (1.50 g) was loaded into a 20 mL glass vial and heated in a water bath at 45 °C for 1 h. Firstly, the instrument was cleaned at a 400 mL min^−1^ flow rate for 1 h to ensure that the aroma profile was round before starting to analyze the samples. The sensor parameters were set to an injection flow rate of 400 mL min^−1^, and the analysis time was 60 s. Subsequently, the syringe was cleaned at a flow rate of 400 mL min^−1^ for 120 s, and then the next sample was measured.

### 2.4. Aroma Extraction by Headspace Solid-Phase Microextraction (HS-SPME)

The YP samples (1.50 g) were added to 3 mL of purified water, transferred into a 20 mL SPME extraction bottle, and mixed thoroughly. The YP samples were heated and equilibrated in a water bath at 45 °C for 20 min. Then, 10 μL of 2-methyl-3-heptanone (0.100 mg/mL in methanol) was added as an internal standard. Subsequently, headspace aroma extraction was performed using a 65 μm DVB/PDMS fiber at 45 °C for 40 min. The extraction fiber was inserted into the injection port at 250 °C for 10 min before use. The loaded fiber was desorbed at the injection port for 5 min using the pulse splitless mode.

### 2.5. GC-MS and GC-MS/O

The qualitative and quantitative analysis of aroma compounds from different YPs was performed using an Agilent 8890 GC equipped with a 5977B mass-selective detector (Agilent Technologies, Santa Clara, CA, USA). The GC–MS was equipped with a sniffing port (ODP3, Gerstel, Germany), and GC effluent was injected with a 1:1 split between the MS and the sniff port for GC-MS/O. The volatile compounds were separated on a DB-WAX column (30 m × 0.25 mm, inner diameter 0.25 μm). Ultra-pure helium (99.999%) was used as the carrier gas at a flow rate of 1.00 mL/min. The pulse splitless mode was used in this work. GC-MS and GC-MS/O of the YPs were conducted according to our previous work, with some modifications. The oven temperature program was as follows: the initial temperature was kept at 35 °C for 1 min, then increased to 100 °C at 4 °C/min and held for 1 min, then increased to 170 °C at 2 °C/min and held for 1 min, and finally, increased to 220 °C at 5 °C/min speed and held for 1 min.

### 2.6. Qualitative and Quantitative Methods

The GC-MS data were analyzed by Qualitative Analysis 10.0 software (Agilent Technologies, Santa Clara, CA, USA), and compared to the NIST20 database. The compounds were qualitatively compared with the MS database, retention index (RI), standard substances, and aroma characteristics. The RI was calculated based on Equation (1).
(1)$$RI=100\times \left[\frac{n+\left(t_{\left(x\right)}\right)-t_{\left(i\right)}}{t_{\left(i+1\right)}-t_{\left(i\right)}}\right]$$

In Equation (1): t*_(x)_* is the retention time of compound *x*; n is the number of n-alkane carbon atoms of the peak before the retention time of compound *x*; t*_(i)_* and t_(*i*+1)_ are the retention times of the n-alkanes with *i* and *i* + 1 carbon atoms, respectively.

The concentration of each volatile compound was calculated as the ratio of the peak area to the internal standard (I_S_) peak area [20]. The concentration of each volatile compound was calculated based on Equation (2).
(2)$$Concentration=\frac{\left(\frac{V_{s}}{I_{s}}\right)\times C\times V}{m}$$

In Equation (2): V_S_ is the peak area of volatile compounds; I_S_ is the peak area of the internal standard, c (mg/mL) is the concentration of the internal standard, V (μL) is the volume of the internal standard, m (g) is the mass of the sample.

### 2.7. Addition Experiment

Addition tests were conducted to compare the aroma profile changes after the potential aroma-active compounds screened from the GC-O and PLSR analyses were added to 1.50 g of YP. All chemical solutions were prepared in water and added to the YP matrix at a volume of 10 µL [18]. The amount of each volatile compound added to the YP matrix was based on the maximum concentration determined in the five YPs [21].

### 2.8. Statistical Analysis

All data were obtained in triplicate, and the results are expressed as the mean ± standard (mean ± SD). The data were statistically analyzed using SPSS Statistics 24 software with ANOVA and Duncan’s test (*p* < 0.05). Partial least squares regression (PLSR) was conducted to explore the correlations among the samples, aroma compounds, and sensory evaluations using XLSTAT. With volatile compounds designated as independent variables (*X*) and sensory evaluation attributes and samples as dependent variables, the importance of the variables in projection (VIP) scores in the context of PLSR reflects the relative importance of each *X* variable to each *Y* variable in the prediction model. The variables with a VIP value higher than 1 were considered to be of potential interest [21].

## 3. Results and Discussion

### 3.1. Sensory Evaluation

Quantitative descriptive sensory analysis was performed to obtain the overall aroma profiles of the YPs (Figure 1A). The results show that YP #10 was significantly different from the other YPs, and the most intense aroma profile in YP #10 was sweaty (*p* < 0.001), followed by rice bran, sour, plastic, floral, roasty, and sweet. However, the rest of the attributes were not obvious variances. The hierarchical clustering results (Figure 1B) indicate that the five YPs could be divided into two categories: cluster 1 (YP #10) and cluster 2 (YPs #11, #12, #13, and #14). The results suggest that the aroma of YP #10 was significantly different from the four other YPs.

### 3.2. Sensor Array Response to YPs

The spatial distribution and distances of the different YPs were analyzed by principal component analysis, and the results are shown in Figure 2. PC1 and PC2 made up 97.43% and 1.85% of the variance, respectively, with a cumulative contribution greater than 90%, indicating that they reflect the information of the overall data characteristics of the samples. The five sample groups were well distinguished by the E-nose results, indicating that these five samples could be completely separated. This result also elucidates that the E-nose could efficiently and accurately distinguish the differences among the variance samples based on their volatile properties [22]. YP #10 is located on the rightmost side of the graph and the farthest away from the other YPs, indicating that the aroma profile of YP #10 was different from the other YPs.

### 3.3. Volatile Compound Analysis

A total of 67 aroma compounds, including alcohols, aldehydes, ketones, esters, acids, hydrocarbons, and others, in five YPs were detected by GC-MS (Figure 3). As the results show, totals of 36, 52, 50, 46, and 46 volatile compounds were detected in YP #10, #11, #12, #13, and #14, respectively. Among them, 26 aroma-active compounds were identified by GC-MS/O, including 8 aldehydes, 8 alcohols, 4 ketones, 2 acids, and 4 others (Table 2).

A total of eight aldehydes were identified in the five YPs. Aldehydes have a lower odor threshold and play a crucial role in food aroma. Among them, hexanal, heptanal, octanal, nonanal, furfural, and benzaldehyde all exist simultaneously in the five YPs, and hexanal, heptanal, and octanal provide a green aroma in the YPs. Additionally, heptanal and benzaldehyde have previously been reported to be important aroma compounds in bread crumb [1]. Studies have shown that these three compounds contribute to rancid or stale aromas that appear after meat products are stored at high temperatures for 48 h [23]. Most aldehydes contribute to unpleasant aroma profiles [24], which also confirms their contribution to the off-odorant in YPs. Octanal may be produced by fatty acid oxidation and thermal degradation [25]. Furfural and benzaldehyde provide a nutty odor in YPs. Furfural is mainly produced by the Maillard reaction, the direct degradation of pentoses, and the indirect conversion of pentosans [26], and is used as one of the indicators of beer color and aroma deterioration [27]. Therefore, the presence of furfural may also be one of the reasons for the off-odorant of YPs. Benzaldehyde is mainly produced by the degradation of phenylalanine [28]. Benzeneacetaldehyde was only detected in YP #10, indicating that benzeneacetaldehyde is an aromatic compound that contributes the floral note in YP #10.

A total of eight kinds of alcohols were detected in the five YPs. Alcohols have a low boiling point and volatile physical and chemical properties. Therefore, they carry less volatile and high-boiling molecules from other ingredients through the volatilization process to enhance aroma [29]. Among them, 1-hexanol, benzyl alcohol, and phenylethyl alcohol were all detected in the five YPs. These three compounds make great contributions to the floral notes of the YPs. 1-Pentanol was not detected in YP #10. Studies have shown that it is related to the fatty odor in chicken soup, with a significant positive correlation [30]; meanwhile, it offers a yeasty odor, which may contribute to the off-odorant in YPs #11, #12, #13, and #14. Furfuryl alcohol only appeared in YP #11 and #12, which is most common in roasty coffee [31]. It can provide a burnt aroma, which may be one of the reasons for the thick roasty aroma in YPs #11 and #12. (3S)-3,7-dimethyloct-7-en-1-ol made a great contribution to the floral aromas of YPs #10, #11, and #12, which may be one of the reasons for their higher floral fragrance scores in the sensory evaluation.

A total of four ketones were detected in the five YPs. Ketones are mainly derived from the oxidative degradation of unsaturated fatty acids, with a large threshold and little contribution to the aroma [30]. 2-Nonanone and 6-methyl-5-hepten-2-one both existed in the five YPs. 2-Nonanone seems to be formed by the β-oxidation of fatty acids in YPs [32], and 6-methyl-5-hepten-2-one is considered to be an off-odorant compound, which may be produced by the Maillard reaction [33].

Three acids only existed in YP #10, providing certain fermented and rancid odors, and they may be one of the sources of the sweaty odor of YP #10. It was confirmed that the three acids detected were butanoic acid, corresponding to an acidic sour odor, 2-methyl-propanoic acid, corresponding to a sweaty odor, and 3-methyl-butanoic acid, corresponding to a cheesy odor. This is also the reason the aroma of YP #10 was different from the other YPs. 3-Methyl-butanoic acid can be degraded by Strecker or produced by the microbial metabolism of the Ehrlich pathway [34]. Some researchers have shown that these acids are also a possible cause of the off-odorants of yeast extract [14].

Sulfur-containing compounds have a significant impact on the overall aroma of YPs due to their extremely low threshold. Dimethyl disulfide was detected in YPs #11, #12, and #14, which had the aroma of sulfide, such as in rotten cabbage. Although present in very low concentrations, this sulfur compound plays a vital role in food aroma [35]. At the same time, in the electronic nose, we found that the sensor had a certain response to sulfur compounds, but few sulfur compounds were detected due to their lower concentrations. A light-induced off-odorant in skim milk has been shown to be a cause, mainly formed by the oxidation of methionine by singlet oxygen [36]. o-Cresol existed in the five kinds of YPs, providing plastic aromas. In red wine, o-cresol was identified as an off-aroma substance [37]. A certain amount of indole was also detected in YPs #10, #11, and #12. Indole has a strong feces odor at high concentrations, but it has a fragrance at lower concentrations [38]. By observing the content of indole, it was found that in YPs #10, #11, and #12, it gradually decreased, which may be one of the reasons for the higher sweaty concentration in YP #10.

### 3.4. Relationship between Volatile Compounds and Sensory Evaluation

To determine the volatile compounds related to the sensory evaluation of the five YPs, a correlation analysis was performed using PLSR, and the results are shown in Figure 3. Most of the volatile compounds are located in the ellipse, R^2^ = 100%, indicating that they can be well explained by the PLSR model. The five YP samples could be divided into two groups according to dimension 1. YP #10 is located on the positive axis of dimension 1, and YPs #11, #12, #13, and #14 are located on the negative axis of dimension 1, which is consistent with the results of the sensory evaluation. Of these 19 volatile compounds, 10 compounds had variable importance in the projected value (VIP ≥ 1), indicating that these 10 compounds significantly contributed to the aromas of the YP samples.

Then, according to the standard correlation coefficient results of the aroma compounds and YP sensory properties, the relationships between the potent aroma active compounds and the sensory properties were predicted (Figure 4). The results show that 2-methyl-propanoic acid, butanoic acid, 3-methyl-butanoic acid, (3S)-3,7-dimethyloct-7-en-1-ol, benzyl alcohol, and indole were strongly positively correlated with the sour attribute. 2-Methyl-propanoic acid, butanoic acid, 3-methyl-butanoic acid, (3S)-3,7-dimethyloct-7-en-1-ol, and indole were strongly positively correlated with the rice bran and sweaty attributes. Heptanal, octanal, and indole were strongly positively correlated with the floral attribute. Hexanal, heptanal, octanal, 2-methyl-propanoic acid, benzyl alcohol, and o-cresol were strongly positively correlated with the sweet attribute. Hexanal, heptanal, octanal, and o-cresol were strongly positively correlated with the roasty attribute. Octanal, 2-methyl-propanoic acid, butanoic acid, 3-methyl-butanoic acid, (3S)-3,7-dimethyloct-7-en-1-ol, and indole were strongly positively correlated with the plastic attribute.

### 3.5. Aroma Addition Experiment Analyst

To verify that all compounds contributing to the overall aroma were correctly identified and quantified, aroma addition experiments were performed. All odorants with a VIP ≥ 1 in different YPs were added to the YP matrix at the highest concentration measured in the YP. The YPs were sensory evaluated by a trained sensory panel. The comparison was based on their similarity on a scale from 1 to 7 and the strength of the selected odor attributes. After adding 10 compounds with a VIP ≥ 1, the sour, plastic, and roasty attributes were all enhanced, and the rice bran attribute was decreased. YP #14 had the most obvious change in all characters. Combined with the results of the PLSR model in Figure 3B it can be found that the compounds that were positively correlated with the sour, plastic, and roasty attributes had stronger effects. These compounds had a certain masking effect on the odor produced by other compounds, which could significantly enhance the functions of these attributes. However, hexanal, o-cresol, and benzyl alcohol were strongly negatively correlated with the rice bran attribute, which could offset the effect of the positively correlated compounds.

## 4. Conclusions

Five YPs exhibited significant differences in their aroma profiles due to variances in their manufacturing processes. The sensory evaluation results show that YP #10, with strong sweaty and rice bran aroma attributes, was significantly different from the other YPs. The rice bran aroma intensity of the other YPs was the highest, followed by the sweaty, sweet, roasty, floral, plastic, and sour aromas. The electronic nose analysis results are consistent with the sensory evaluation results, which can further distinguish the aroma differences among the five YPs. A total of 27 aroma-active compounds were identified by SPME-GC-MS/O in the 5 YPs, including 8 aldehydes, 8 alcohols, 4 ketones, 3 acids, and 4 others. The PLSR results show that o-cresol, (3S)-3,7-dimethyloct-7-en-1-ol, benzyl alcohol, octanal, 2-methyl-propanoic acid, butanoic acid, 3-methyl-butanoic acid, hexanal, heptanal, and indole with VIP values ≥ 1 were predicted as the potential aroma compounds. Addition tests further confirmed that the sour, plastic, and roasty attributes were all enhanced, and only the rice bran attributes were decreased. Moreover, the overall aroma profiles showed expanding trends, elucidating the important contributions of these ten aroma-active compounds to YPs. This work compared the key aroma compounds of different YPs by GC-MS/O combined with addition tests for the first time, which provides guidance for aroma quality improvement and expanding the application of YPs.

## Figures and Tables

**Figure 1 foods-12-03136-f001:**
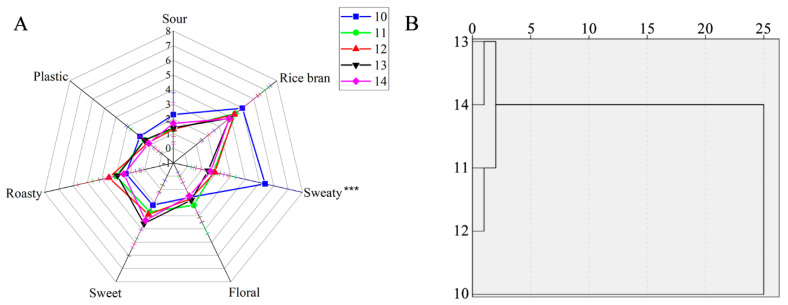
Sensory evaluation results of yeast protein samples ((**A**) radar chart of quantitative descriptive analysis; (**B**) hierarchical clustering analysis of five yeast protein samples). (*** indicates *p* < 0.001).

**Figure 2 foods-12-03136-f002:**
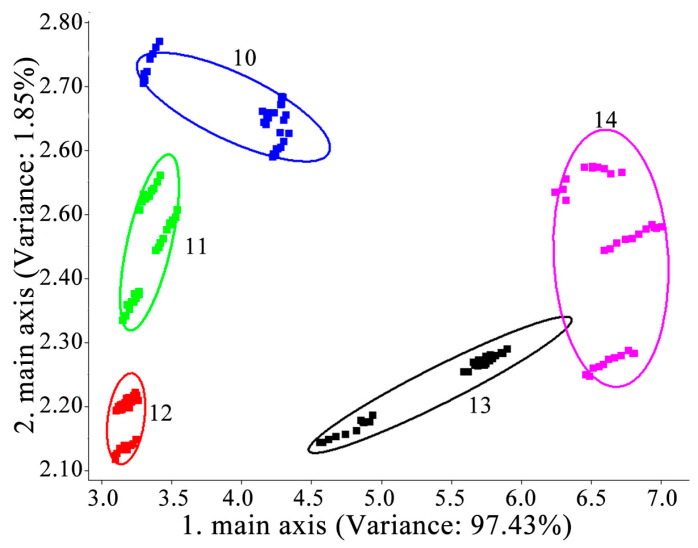
Principal component analysis results of electronic nose of five yeast proteins samples.

**Figure 3 foods-12-03136-f003:**
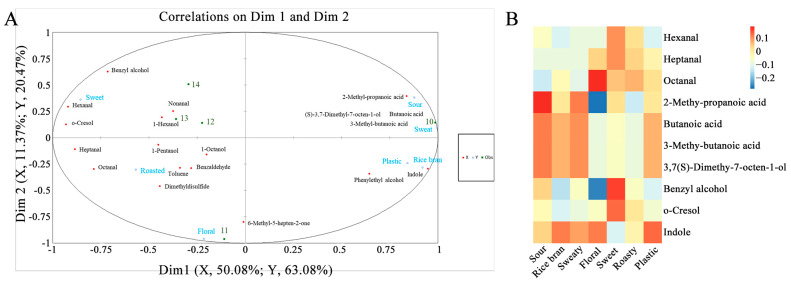
The correlation matrix of the sensory attributes to the volatile compounds among five yeast proteins. ((**A**) The correlation matrix of the aroma-active compounds to sensory attributes. The red plots represent the 19 aroma-active compounds with GC-O. The green plots represent the 5 yeast protein samples. The blue circles represent the 7 aroma attributes. (**B**) Heat map of standard correlation coefficients of the aroma-active compounds to the aroma profiles).

**Figure 4 foods-12-03136-f004:**
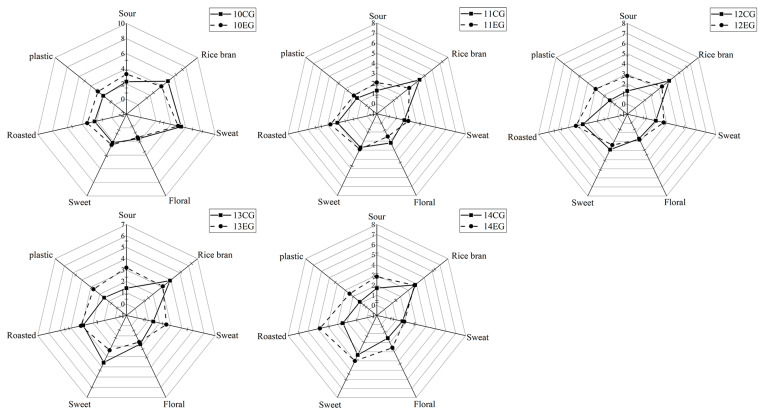
Comparative aroma profile analysis of the yeast protein group (solid line) and the corresponding aroma addition group (dashed line).

**Table 1 foods-12-03136-t001:** Sensor characteristics of the 10-sensor array in the electronic nose.

Array	Sensor Types	Properties of Different Arrays
S1	W1C	Aromatic compounds, benzene
S2	W5S	Sensitive to nitrogen oxides
S3	W3C	Ammonia, aromatic compounds
S4	W6S	Selective mainly for hydrogen
S5	W5C	Alkanes, aromatic compounds
S6	W1S	Short-chain alkanes, such as methane
S7	W1W	Sulfur organic compound
S8	W2S	Sensitive to alcohols, ethers, aldehydes, and ketones
S9	W2W	Aromatic compounds, sulfur organic compounds
S10	W3S	Sensitive to long-chain alkanes

**Table 2 foods-12-03136-t002:** Identification and quantification of aroma-active compounds in different yeast proteins by SPME-GC-MS-O.

No.	RI ^a^	Compounds	CAS	Relative Concentration (μg/kg) ^b^					Aroma Quality
				#10	#11	#12	#13	#14	
		Aldehydes							
1	1089/1081	Hexanal	66-25-1	59.06 ± 17.68 ^b^	129.25 ± 38.98 ^a^	195.11 ± 86.30 ^a^	186.40 ± 45.36 ^a^	173.49 ± 11.42 ^a^	Green, grass
2	1174/1181	Heptanal	111-71-7	76.75 ± 3.88 ^c^	935.69 ± 76.68 ^a^	557.69 ± 98.11 ^b^	1176.58 ± 325.61 ^b^	983.97 ± 112.90 ^a^	Citrus, green
3	1270/1284	Octanal	124-13-0	48.34 ± 18.40 ^c^	248.47 ± 55.19 ^a^	164.89 ± 28.62 ^b^	311.21 ± 47.77 ^a^	179.78 ± 69.25 ^b^	Citrus, fat, green
4	1388/1368	Nonanal	124-19-6	94.94 ± 31.36 ^b^	361.59 ± 38.03 ^a^	161.99 ± 96.95 ^b^	374.64 ± 58.93 ^a^	306.62 ± 66.73 ^a^	Green, lemon
5	1458/1450	Furfural	98-01-1	10.20 ± 6.24 ^b^	8.05 ± 3.67 ^b^	8.01 ± 1.66 ^b^	16.97 ± 8.45 ^a^	19.55 ± 4.14 ^a^	Baked potatoes, bread
6	1503/1504	Benzaldehyde	100-52-7	940.80 ± 391.75 ^b^	2230.07 ± 389.90 ^a^	1695.19 ± 671.23 ^b^	3042.13 ± 1297.22 ^a^	2287.81 ± 177.01 ^a^	Bitter almond, green
7	1528/1523	(*Z*)-2-Nonenal	60784-31-8		1.37 ± 0.00 ^d^	2.84 ± 0.00 ^a^	1.79 ± 0.00 ^b^	1.64 ± 0.06 ^c^	Green, fatty
8	1667/1625	Benzeneacetaldehyde	122-78-1	11.99 ± 7.70					Floral, honey, sweet
		Alcohols							
9	1249/1261	1-Pentanol	71-41-0		9.13 ± 1.06 ^b^	4.17 ± 0.12 ^c^	14.86 ± 3.33 ^a^	13.17 ± 1.99 ^a^	Fruit, solvent, sweet
10	1349/1345	1-Hexanol	111-27-3	10.83 ± 3.19 ^c^	37.74 ± 2.41 ^b^	27.50 ± 6.69 ^b^	54.37 ± 16.52 ^a^	61.03 ± 5.25 ^a^	Grassy, marzipan-like
11	1455/1454	1-Heptanol	111-70-6		35.45 ± 26.09 ^b^	117.34 ± 43.38 ^a^	74.94 ± 31.79 ^a^	61.84 ± 3.38 ^b^	Green, fatty
12	1564/1558	1-Octanol	111-87-5	16.09 ± 6.12 ^b^	50.62 ± 9.69 ^a^	48.05 ± 19.79 ^a^	66.62 ± 29.42 ^a^	59.19 ± 3.89 ^a^	Creamy, sweet
13	1640/1656	Furfuryl alcohol	98-00-0		8.38 ± 0.00	7.35 ± 2.07			Bread, sweet, roasty
14	1767/1767	(3S)-3,7-Dimethyloct-7-en-1-ol	6812-78-8	26.37 ± 9.79 ^a^	6.28 ± 1.13 ^b^	2.17 ± 0.93 ^b^			Floral, sweet
15	1879/1863	Benzyl alcohol	100-51-6	6.34 ± 2.51 ^b^	10.90 ± 0.75 ^a^	9.52 ± 3.87 ^a^	13.28 ± 6.25 ^a^	9.78 ± 1.06 ^a^	Sweet, floral
16	1873/1895	Phenylethyl Alcohol	60-12-8	25.34 ± 6.41 ^a^	26.43 ± 0.98 ^a^	5.16 ± 2.41 ^b^	26.54 ± 12.31 ^a^	11.38 ± 1.27 ^b^	Fruit, honey, sweet
		Ketones							
17	1283/1281	2-Octanone	111-13-7			187.80 ± 136.86 ^a^	64.20 ± 0.00 ^b^	34.04 ± 0.38 ^b^	Fat, fragrant, mold
18	1365/1325	6-Methyl-5-hepten-2-one	110-93-0	19.33 ± 7.49 ^b^	76.58 ± 2.03 ^b^	52.23 ± 21.59 ^b^	154.37 ± 39.67 ^a^	157.69 ± 34.33 ^a^	Citrus, sweet, strawberry
19	1387/1364	2-Nonanone	821-55-6	8.71 ± 1.21 ^b^	15.30 ± 7.60 ^b^	92.55 ± 60.54 ^a^	27.99 ± 6.58 ^b^	14.84 ± 7.56 ^b^	Fruit, green
20	1489/1479	2-Decanone	693-54-9		4.55 ± 0.57 ^b^	39.40 ± 26.59 ^a^	6.75 ± 2.03 ^b^	5.62 ± 2.00 ^b^	Fat, fruity
		Acids							
21	1557/1631	2-Methyl-propanoic acid	79-31-2	6.37 ± 4.99					Sour, butter, sweat
22	1630/1685	Butanoic acid	107-92-6	47.46 ± 33.10					Butter, cheese, sour
23	1633/1694	3-Methyl-butanoic acid	503-74-2	22.50 ± 15.50					Sweaty, cheese
		Others							
24	1146/1136	3-Carene		14.49 ± 1.00 ^b^	22.86 ± 6.49 ^a^	13.75 ± 1.45 ^b^	31.49 ± 9.35 ^a^		Woody
25	1071/1071	Dimethyl disulfide	624-92-0		9.87 ± 7.80	10.40 ± 10.73		5.54 ± 3.15	Cabbage, garlic, onion
26	1934/1990	o-Cresol	95-48-7	4.79 ± 2.36 ^c^	53.51 ± 3.76 ^a^	48.59 ± 19.35 ^b^	89.40 ± 40.25 ^a^	76.07 ± 7.59 ^a^	Musty, plastic, medicinal
27	2412/2404	Indole	120-72-9	943.40 ± 386.62 ^a^	325.33 ± 35.15 ^b^	4.46 ± 4.65 ^c^			Burnt, mothball

(a) RI value is from the literature/calculation; (b) #10, #11, #12, #13, and #14 represent the abbreviations of 5 yeast protein samples. abcd is a significant difference analysis of the data in the table without annotation.

## Data Availability

The datasets generated for this study are available on request to the corresponding author.

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
