# Peer review of "Characterization of the Key Aroma Compounds in Different Yeast Proteins by GC-MS/O, Sensory Evaluation, and E-Nose"

_foods, 2023, doi:10.3390/foods12163136_

Round 1

Reviewer 1 Report

Comments and Suggestions for Authors

            The title of manuscript is “Characterization of the key aroma compounds in different yeast proteins by GC-MS/O, sensory evaluation and E-nose”. The aim of the work were to:

(1) determine the aroma profile of yeast proteins by sensory evaluation and electronic nose analysis,

(2) identify the aroma-actives compounds in yeast proteins by headspace SPME-GC-MS/O,

(3) correlate the relationship between the aroma-active compounds to the aroma profiles and screen the potential aroma compounds,

(4) confirm the key aroma-active compounds and their contributions by addition  tests.

I commented on the manuscript and the comments are presented below:

The Introduction of the study provides with some general information about the used techniques of the odor measurement. On the other hand, for several decades, research has been carried out on the use of various types of odor measurement techniques. I suggest supplementing the Chapter with additional information related to research problem. Additional information contained in the Introduction chapter will make the aim of the study will clearly stated. I suggest supplementing the Chapter with additional information, for example: “The aroma profile of wheat bread crumb influenced by yeast concentration and fermentation temperature”, “Detection and measurement of aroma compounds with the electronic nose and a novel method for MOS sensor signal analysis during the wheat bread making process”, “Evolution of volatile compounds in gluten-free bread: From dough to crumb”, “Impact of Commercial Inactive Yeast Derivatives on Antiradical Properties, Volatile and Sensorial Profiles of Grašac Wines”

The Materials and Methods section provides the reader with enough information to repeat the experiments conducted. The Authors did not specify how the signal from the sensors used, which was their response to the presence of odor compounds, was analyzed? This should be explained.

In the Results and discussion chapter contains information should be supplemented on discussing with the other items from the last years of publication including investigated problem. I suggest supplementing the Chapter with additional information, for example: “The aroma profile of wheat bread crumb influenced by yeast concentration and fermentation temperature”, “Detection and measurement of aroma compounds with the electronic nose and a novel method for MOS sensor signal analysis during the wheat bread making process”, “Non-Conventional Yeast: Behavior under Pure Culture, Sequential and Aeration Conditions in Beer Fermentation”, “Diversity of Volatile Aroma Compound Composition Produced by Non-Saccharomyces Yeasts in the Early Phase of Grape Must Fermentation”.

The conclusions are well and were supported by the data.

The literature used is appropriate but should be supplementing about the items from the last years of publication.

Author Response

Thank you for your letter and for the reviewers’ comments concerning our manuscript. Those comments are very valuable and helpful for revising and improving our paper. We read all the comments carefully and made revisions which we hope meet with approval. The point-by-point responses to the reviewers’ comments are appended below. The detailed revisions with “Tracked Changes” are in the manuscript, and most of long sentences deleted or revised are marked red for better review.

Thank you for your patience once more. We are looking forward to your positive response.

Yours sincerely,

Jiahui Chen

Reviewer: 1

Comments:

Q1. I suggest supplementing the Chapter with additional information related to research problem. Additional information contained in the Introduction chapter will make the aim of the study will clearly stated. I suggest supplementing the Chapter with additional information, for example: “The aroma profile of wheat bread crumb influenced by yeast concentration and fermentation temperature”, “Detection and measurement of aroma compounds with the electronic nose and a novel method for MOS sensor signal analysis during the wheat bread making process”, “Evolution of volatile compounds in gluten-free bread: From dough to crumb”, “Impact of Commercial Inactive Yeast Derivatives on Antiradical Properties, Volatile and Sensorial Profiles of Grašac Wines”

Response: Thanks for your suggestion. These supplement data were all added at the introduction section: “During bread preparation the important aroma compounds including 2-phenylethanol, 1-heptanol, heptanal, benzaldehyde, phenylacetaldehyde and ethyl octanoate are generated by yeast fermentation in bread crumbs [1]. Moreover, the diversity of yeast strains can affect the flavor complexity of wine [2]. ”. (Line 30-33)

The means of this work were also be strengthen: “The application/replacement of yeast protein in vegetable protein analog affecting the flavor quality is lack of investigation.”. (Line 86-88)

Q2. The Materials and Methods section provides the reader with enough information to repeat the experiments conducted. The Authors did not specify how the signal from the sensors used, which was their response to the presence of odor compounds, was analyzed? This should be explained.

Response: Thanks, the mechanism/principle of the electronic nose response to the aroma compounds were provided: An electronic nose (E-nose) is an array of gas sensors that can make a fingerprint response to specific volatiles. The volatile molecules react with the sensing material of the gas sensor, resulting in irreversible changes in the electrical correlation properties. These changes can be used by pattern recognition algorithms such as artificial neural networks (ANN) to differentiate and classify the different samples [18]. (Line 121-125)

Q3. In the Results and discussion chapter contains information should be supplemented on discussing with the other items from the last years of publication including investigated problem. I suggest supplementing the Chapter with additional information, for example: “The aroma profile of wheat bread crumb influenced by yeast concentration and fermentation temperature”, “Detection and measurement of aroma compounds with the electronic nose and a novel method for MOS sensor signal analysis during the wheat bread making process”, “Non-Conventional Yeast: Behavior under Pure Culture, Sequential and Aeration Conditions in Beer Fermentation”, “Diversity of Volatile Aroma Compound Composition Produced by Non-Saccharomyces Yeasts in the Early Phase of Grape Must Fermentation”.

Response: Thanks, the result section were improved by adding discussions.

(1) The E-nose part “Five sample groups were well distinguished by E-nose results, indicating that these five samples could be completely separated. This result was also elucidated that the E-nose could efficiently and accurately distinguish the differences among variances sample based on their volatile property[24].” (Line 208-211)

(2) The aroma composition of YPs: “Besides, heptanal and benzaldehyde have previously been reported to be important aroma compounds in bread crumb [1].”(Line 231-233); “Most of aldehydes contributed to the unpleasure aroma profiles [24], which also confirmed their contribution to the off-odorant in YP.” (Line 235-236)

Q4. The conclusions are well and were supported by the data.

The literature used is appropriate but should be supplementing about the items from the last years of publication.

Response: Thanks, several latest references were added.

  1. Gancarz, M., Malaga-Toboła, U., Oniszczuk, A., Tabor, S., Oniszczuk, T., Gawrysiak-Witulska, M., & Rusinek, R. Detection and measurement of aroma compounds with the electronic nose and a novel method for MOS sensor signal analysis during the wheat bread making process. Food and Bioproducts Processing, 2021, 127, 90-98.
  2. Pu, D., Shan, Y., Zhang, L., Sun, B., & Zhang, Y. Identification and inhibition of the key off-odorants in duck broth by means of the sensomics approach and binary odor mixture. Journal of Agricultural and Food Chemistry, 2022, 70(41), 13367-13378.
  3. Zeng, L., Fu, Y., Liu, Y., Huang, J., Chen, J, Yin, J, Jin S., Sun W., Xu, Y. Comparative analysis of different grades of Tieguanyin oolong tea based on metabolomics and sensory evaluation. LWT, 2023, 174, 114423.
  4. Li, R., Chen, K., Li, L., Zhao, S., Guo, C., Wang, X., Zhang, J.,Liang, M. Identification of key odor compounds in the burnt smell of upper tobacco leaves through the molecular sensory science technique. Science Asia, 2023, 49(2), 290-296.

Reviewer 2 Report

Comments and Suggestions for Authors

A few suggestions: 

- The introduction is unable to explain why this study is being carried out and why it would be important to do so.

- could you please explain in deep the sampling phase?

- on some samples there are very few replicas especially as regards the measurements made with E-nose I think that at least 100/150 replicas are needed to understand the typical variability of the sample

- for the GC at least 5 replicas are needed

It is not an innovative study but certainly the biggest problem is related to the small number of replicas made, I recommend adding replicas to make the data scientifically more solid

Comments on the Quality of English Language

I suggest to read the text more carefully

Author Response

Thank you for your letter and for the reviewers’ comments concerning our manuscript. Those comments are very valuable and helpful for revising and improving our paper. We read all the comments carefully and made revisions which we hope meet with approval. The point-by-point responses to the reviewers’ comments are appended below. The detailed revisions with “Tracked Changes” are in the manuscript, and most of long sentences deleted or revised are marked red for better review.

Thank you for your patience once more. We are looking forward to your positive response.

Yours sincerely,

Jiahui Chen

Reviewer: 2

Comments:

Q1. The introduction is unable to explain why this study is being carried out and why it would be important to do so.

Response: The microbial protein has low energy consumption, low pollution, short preparation cycle, and balanced nutrition to meet current needs. Compared with plant protein, yeast based alternatives are more nutritious. However, the application/replacement of yeast protein in vegetable protein meat affected the flavor quality is lack of investigation. Therefore, in this work, identification of the key aroma compounds in YP samples is worth of investigation. (Line 86-90)

Q2. could you please explain in deep the sampling phase?

Response: the detailed procedures of yeast protein were provided: “Yeast protein is produced by two steps: firstly, yeast cell walls undergo self-dissolution and fragmentation by adding enzymes; secondly, then the Yeast protein were obtained by centrifugation, dispersion and drying.” (Line 98-100)

Q3. on some samples there are very few replicas especially as regards the measurements made with E-nose I think that at least 100/150 replicas are needed to understand the typical variability of the sample.

Response: Thanks for your suggestion, we have increased the YP samples (n=30) for E-nose analysis. Currently, these samples could well distinguish the variances among five groups of YP. (Line 212-217)

.

Figure 2 Principal component analysis results of electronic nose of five yeast proteins samples.

Q4. for the GC at least 5 replicas are needed.

Response: In this work the aroma extraction and quantification were conducted by SPME-GC-MS, and the aroma compounds were determined by semi-quantification methods. All of the samples were repeated in triplicates. The relative standard deviation of the added internal standard was lower than 10%. These results suggested that three parallel SPME qualification was acceptable. For example, many literature conducted three times in SPME-GC-MS analysis:

[1] Zhou, X., Chong, Y., Ding, Y., Gu, S., & Liu, L. (2016). Determination of the effects of different washing processes on aroma characteristics in silver carp mince by MMSE–GC–MS, e-nose and sensory evaluation. Food chemistry, 207, 205-213.

[2] Zianni, R., Mentana, A., Tomaiuolo, M., Campaniello, M., Iammarino, M., Centonze, D., & Palermo, C. (2023). Volatolomic approach by HS-SPME/GC–MS and chemometric evaluations [3] for the discrimination of X-ray irradiated mozzarella cheese. Food Chemistry, 423, 136239.

[3] Wang, S., He, Y. U., Wang, Y., Tao, N., Wu, X., Wang, X., ... & Ma, M. (2016). Comparison of flavour qualities of three sourced Eriocheir sinensis. Food Chemistry, 200, 24-31.
